# Spatio-Temporal Assessment of Heavy-Duty Truck Incident and Inspection Data

Amy Moore * , Vivek Sujan , Adam Siekmann , Hyeonsup Lim , Shiqi (Shawn) Ou and Sarah Tennille

Oak Ridge National Laboratory, Oak Ridge, TN 37830, USA; sujanva@ornl.gov (V.S.); siekmanna@ornl.gov (A.S.); limh@ornl.gov (H.L.); ous1@ornl.gov (S.O.); bleakneysa@ornl.gov (S.T.)
* Correspondence: mooream@ornl.gov

**Abstract:** Vehicular incidents, especially those involving tractor trailers, are increasing in number every year. These events are extremely costly for fleets, in terms of damage or loss of property, loss of efficiency, and certainly in terms of loss of life. Although the U.S. Department of Transportation (DOT) is responsible for performing inspections, and fleet managers are encouraged to maintain their fleet and participate in regular inspections, it is uncertain whether these inspections are occurring at a frequency that is necessary to prevent incidents. The Federal Motor Carrier Safety Administration (FMCSA) of the DOT manages and maintains the Motor Carrier Management Information System (MCMIS) dataset, which contains all incident and inspection data regarding commercial vehicles in the U.S. The purpose of this preliminary analysis was to explore the MCMIS dataset through spatiotemporal analyses, to uncover findings that may hint at potential improvements in the DOT inspection process and highlight location-specific trends in the dataset. These analyses are novel, as previous research using the MCMIS dataset only examined the data at the state or county level, not at a national scale. The results from the analyses pinpointed specific major metropolitan areas, namely Harris County (Houston), Texas, and three of the New York boroughs (Kings, Queens, and the Bronx), which were found to have increasing incident rates during the study period (2016–2020). An overview of potential causal factors contributing to this increase are provided as well as an overview of the inspection process, and suggestions for improvement relative to the highlighted locations in Texas and New York are also provided. Ultimately, it is suggested that the incorporation of advanced technology and automation may prove beneficial in reducing the occurrence of events that lead to incidents and may also help in the inspection process.

**Keywords:** heavy-duty truck; incidents; inspections; MCMIS; FAF



## 1. Introduction

Vehicular incident rates have been on the rise in the U.S. and internationally. In 2020, more than 5.2 million vehicular incidents occurred, which resulted in almost 36,000 deaths [1]. In 2021, fatalities occurring from vehicular incidents increased to over 42,000. It is estimated that injuries sustained in vehicular accidents will be the seventh leading cause of death worldwide [2]. The occurrence of vehicular incidents involving tractor trailers has also increased [3]. In recent years, truck volumes (both heavy-duty and medium-duty) have increased [4] due to increases in shipping needs and the prevalence of ecommerce. Although an increase in ecommerce can result in a reduction in vehicle miles traveled (VMT) [5] the increase in truck volume can lead to a higher occurrence of incidents involving trucks. These incidents are costly: with average health costs and property damage costs associated with truck incidents ranging from USD 300k to USD 1.2 million, depending on the number of trailers, and incidents involving fatalities costing several millions, depending on damages [6]. These incidents are costly for businesses in terms of loss or damage of property, hindrances in efficiency, but most importantly, in terms of lives lost.

Many truck incidents are preventable: while many incidents are caused by speeding, inclement weather, or driver intoxication, the most common cause of incidents is driver fatigue [6], thus prompting federal regulations prohibiting truck drivers from driving for more than eight consecutive hours without a break. However, another main cause cited for both U.S. and international truck incidents is vehicle maintenance [7–9]. In the U.S., the importance of the Department of Transportation (DOT) to conduct regular and adequate vehicle inspections and pinpoint preventable maintenance is crucial in potentially reducing the number of truck incidents. However, truck drivers and fleet managers must abide by DOT guidance, especially when a vehicle is rendered out-of-service due to unaddressed maintenance issues.

The Federal Motor Carrier Safety Administration (FMCSA) of the DOT is responsible for setting guidelines and contracting qualified inspectors to perform truck inspections at designated locations across the U.S., whether at an interstate truck stop or DOT-designated truck shop. During a DOT truck inspection, the driver of the vehicle will be issued a result: no violations, non-out-of-service (Non-OOS) violation (less severe), or out-of-service (OOS) violation (more severe) [10]. During an inspection, a driver can receive multiple violations (there are a total of 50 categories). If a driver receives one OOS violation, the driver is unable to return to operating the vehicle until the issues resulting in the OOS designation are addressed accordingly. In some cases, this can be catastrophic for the driver: with an inability to earn an income until the vehicle is serviced, inspected again, and deemed road worthy. Unfortunately, this can present opportunities for falsification of documents or illegally driving a vehicle although appropriate maintenance has not been performed. Although preventative maintenance is costly up-front, it can be an incentive for drivers and fleet managers to prevent the vehicle from being removed from the road, maintain higher freight energy efficiency, and increase overall asset availability and productivity.

The FMCSA's Motor Carrier Management Information System (MCMIS) dataset, which is available for public download [11], contains all truck (light-, medium-, and heavy-duty) incident and inspection data for every recorded road event in the U.S. (including Alaska, Hawaii, and Puerto Rico). The incident data provides locational and temporal attributes, as well as other attributes detailing the severity and possible causes of incidents involving commercial vehicles. Inspection data is also provided and contains details regarding the violation (if any) type and number for each event.

This paper will provide an in-depth county-level breakdown of reported truck incidents and inspections from 2016 to 2020. Prior to this study, the MCMIS dataset had never been evaluated at a national scale [12,13]. Previous work using the dataset has involved specific, county or state-level analyses. This paper will provide the reader with an overview of trends, anomalies, and expected key findings based on previous research. Ultimately, the purpose of this data exploration was to examine incident and inspection rates relative to locations to see where problems potentially lie to uncover future solutions for the FMCSA, inspectors, fleets, and state-level DOTs. It serves to be a resource for decision makers, namely state-level DOTs, to consider when examining incident and inspection rates, relative to losses in efficiency, and ultimately, safety improvements. This paper is structured as follows. In Section 1, the introduction, the motivating problem, and terminology are presented. In Section 2, this paper will present the MCMIS dataset, used in this analysis as well as justification for the categorization of results found in the analysis. In Section 3, the results and analysis are presented, along with breakdowns at the national and county-level as well as by inspection violation category. Lastly, Section 4 provides concluding remarks and potential directions for expanded research on the topic.

## 2. MCMIS Dataset

The FMCSA manages the MCMIS dataset, which contains detailed information regarding the fitness of on-road trucks, buses, and hazardous material (hazmat) shippers who must follow federal regulations regarding maintenance of these vehicles [14]. For the purposes of this research, only the incident data files, public inspections, and inspection

violation files were obtained directly from the FMCSA for the years 2016 through 2020. Although the 2021 incident data were available, 2021 inspection data were not. Thus, the 2021 data were excluded from the analyses.

Initially, the datasets were broken down to examine total numbers of incidents and inspections per year (see Table 1 and Figure 1) with the assumption that total numbers would be at their lowest in 2020 due to the COVID-19 pandemic. Spatiotemporal analyses were performed to determine incident prevalence during time of day, with the assumption that most would occur during the afternoon [14]. Other factors such as inclement weather, the presence of road construction, and road condition/presence of debris were also considered, as those are known causes of truck incidents. However, it was determined that the focus of this work would be in examining incidents relative to the top violation causes (Table 2).

**Table 1.** Incidents and inspections by year.

| Year | Incidents | Inspections |
|---|---|---|
| 2016 | 178,098 | 3,399,161 |
| 2017 | 182,764 | 3,457,036 |
| 2018 | 196,629 | 3,419,098 |
| 2019 | 194,613 | 3,472,480 |
| 2020 | 167,665 | 2,582,565 |

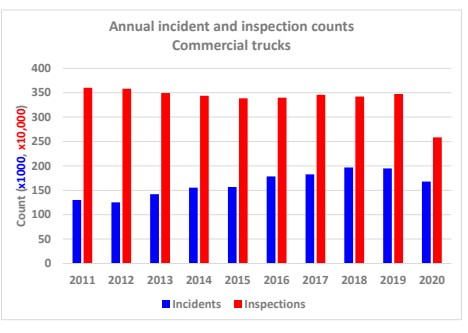 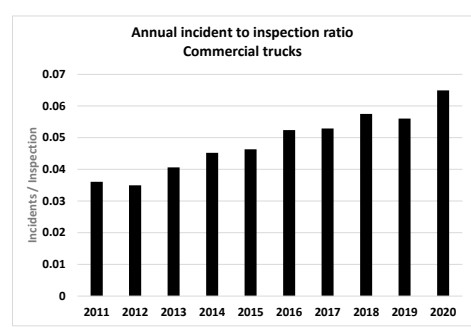

**Figure 1.** Heavy-duty freight truck annual incidents and inspections (**left**), and the associated incident-to-inspection ratio (**right**) for 2011 to 2020.

**Table 2.** OOS and Non-OOS violation categories by year.

| Year | OOS | Non-OOS |
|---|---|---|
| 2016 | Brakes, tires, lighting, load, and brake adjustment | Lighting, vehicle defects, brakes, other driver violations, and tires |
| 2017 | Brakes, tires, lighting, load, and brake adjustment | Lighting, vehicle defects, brakes, other driver violations, and tires |
| 2018 | Brakes, tires, lighting, load, and other driver violations | Lighting, vehicle defects, brakes, other driver violations, and tires |
| 2019 | Brakes, tires, lighting, load, and other driver violations | Lighting, vehicle defects, brakes, other driver violations, and tires |
| 2020 | Brakes, tires, lighting, load, and other driver violations | Lighting, vehicle defects, brakes, other driver violations, and tires |

*Data Processing*

For each of the five years ranging from 2016 to 2020, the incident file was used to obtain the number of incidents per Federal Information Processing System (FIPS) code by U.S. county, as well as to assign a bin based on the time of the day that the incident was reported. The bins used for the assignment were: early (before 4 a.m.), morning (4 a.m. to 10 a.m.), afternoon (10 a.m. to 4 p.m.), evening (4 p.m. to 6 p.m.), and late (10 p.m. to midnight). These time bins were chosen arbitrarily but still represented the assumed distribution, namely the prevalence of incidents in the afternoon time period.

The inspection file was used to obtain the number of inspections (using an inspection ID), along with a designated time bin for each FIPS county code. The inspection violation

file was then joined to the inspection file based on the unique inspection ID to obtain violation category information. Although the incident file was manageable in Microsoft Excel, the analyses involving the inspections and inspection violation files required the use of the Python Pandas library to perform the analyses. The final raw output was in csv form (see headings in Table 3), and the total truck VMT per county per day was used to find normalized values for each variable. The VMT was found using the Freight Analysis Framework (FAF) dataset to obtain truck volumes per network link and the length of each network link (see Figure 2) throughout each of the 3143 U.S. counties [15]. The FAF dataset contains approximately 500,000 road segments, with segment lengths and freight flows (freight-carrying trucks) on these links provided in the attribute table. The process of finding the VMT per county involved obtaining FAF network segments per county and obtaining the product of the freight flow (volume) and total mileage. The result of this process is shown in Figure 3. These data were fed into a Microsoft Excel macro, which processed the data and provided output tables for further analysis.

**Table 3.** Row headings for the raw data file.

| FIPS Inspection ID | Inspections (OOS) | Lighting Violations (Non-OOS) |
|---|---|---|
| State | Inspections (Non-OOS) | Vehicle defect violations (Non-OOS) |
| County | Brake violations (OOS) | Brake violations (Non-OOS) |
| Time Bin | Tire violations (OOS) | Driver violations (Non-OOS) |
| Latitude | Lighting violations (OOS) | Tire violations (Non-OOS) |
| Longitude | Load violations (OOS) | |
| Crashes (total) | Brake adjustment violations (OOS) | |

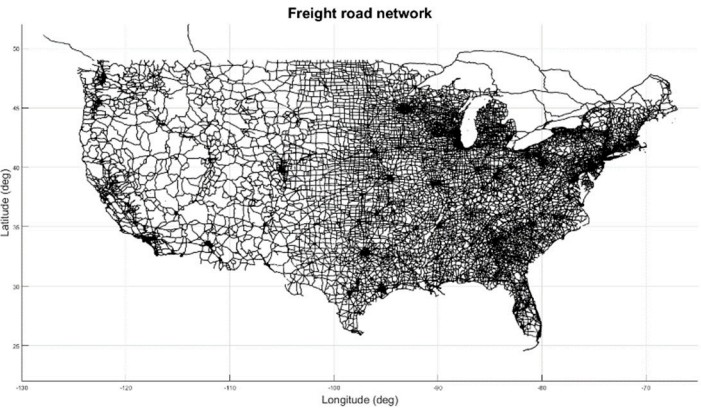

**Figure 2.** Heavy-duty tractor-trailer freight road network across the contiguous United States.

For each of the five years of data, the data were grouped into three categories, which were sorted by county: total inspections, OOS violations, and Non-OOS violations. Scatter plots were used to plot total incidents (Figure 4), where each point represented an individual county. Additional scatter plots also showed the incident-to-inspection ratio, relative to inspections for each county. These plots were grouped into nine boxes, with designations given to each box based on the scenario description (Table 4).

These designations were given based on several assumptions. Firstly, scenarios with low overall incidents were deemed ideal. Second, scenarios deemed less than ideal involved high inspections and moderate and high levels of incidents. The reasoning for this assumed that inspections are costly, and if inspections are high, while incidents are moderate to high, this is inefficient and not cost-effective. However, the scenario with low incidents and high inspections could also be considered ideal, as the high number of inspections could be deemed as preventative. Lastly, the scenarios deemed as not ideal involved high numbers

of incidents, and a moderate number of incidents with low inspections. This last scenario was deemed not ideal due to the assumption that incidents could have been prevented with an increase in inspections.

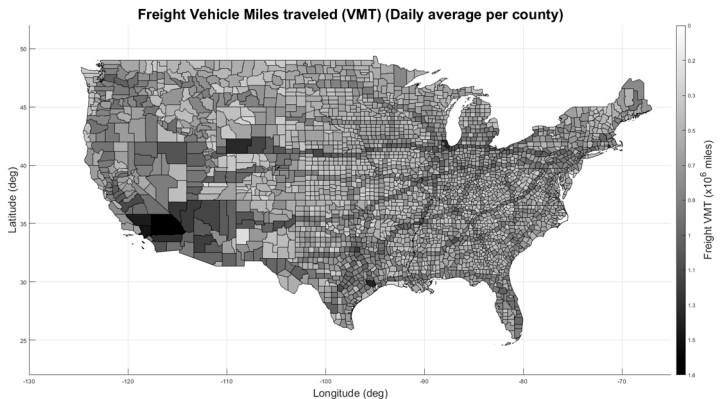

**Figure 3.** Heavy-duty tractor-trailer freight vehicle miles traveled per county (averaged per day).

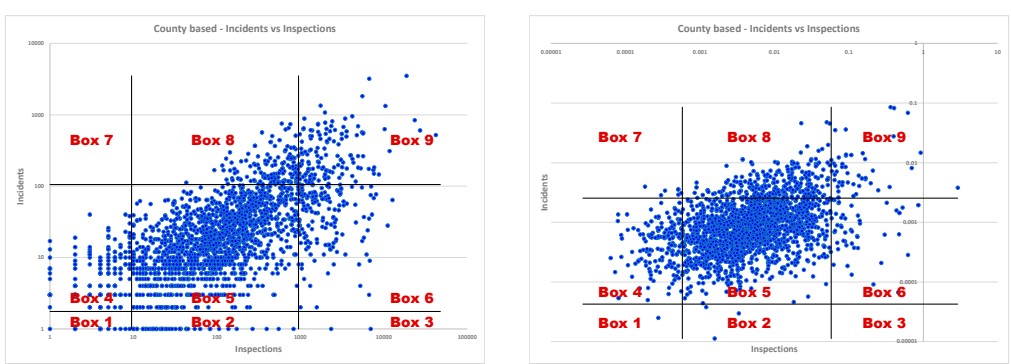

**Figure 4.** Scatterplot of incidents and inspections (absolute values (**left**); normalized by county daily VMT (**right**)) in 2016. Each point represents a single county.

**Table 4.** Description of the nine boxes.

| Description | Designation | Description | Designation | Description | Designation |
|---|---|---|---|---|---|
| Box 7 | | Box 8 | | Box 9 | |
| High incidents (incidents/inspection ratio), low inspections | Not ideal | High incidents (incidents/inspection ratio), moderate inspections | Not ideal | High incidents (incidents/inspection ratio), high inspections | Not ideal |
| Box 4 | | Box 5 | | Box 6 | |
| Moderate incidents (incidents/inspection ratio), low inspections | Not ideal | Moderate incidents (incidents/inspection ratio), moderate inspections | Less than ideal | Moderate incidents (incidents/inspection ratio), high inspections | Less than ideal |
| Box 1 | | Box 2 | | Box 3 | |
| Low incidents (incidents/inspection ratio), low inspections | Ideal | Low incidents (incidents/inspection ratio), moderate inspections | Ideal | Low incidents (incidents/inspection ratio), high inspections | Less than ideal |

## 3. Results and Analysis

### 3.1. Trends across the Nation

For this analysis, simple assumptions were made based on previous research examining trends in truck incidents across the U.S. Some of these assumptions were confirmed through examination of the MCMIS dataset. From Figure 5, it is apparent that 2016 was a typical year for incident rates. Figure 5 also shows that, for 2016, most incidents occurred in the afternoon. In Figure 6, the assumption was that, in 2016, incidents would occur during peak travel months and months when shipping frequency increases, which coincides with major holidays, and this was confirmed, as the greatest spike in incidents in 2016 occurred in December. It was also assumed that, prior to normalizing the data, states with higher populations would experience higher rates of incidents simply due to the presence of higher traffic volumes. This is apparent in Figure 7, where California, Texas, and Florida all stand out, as these states had the highest populations in the U.S. in 2016.

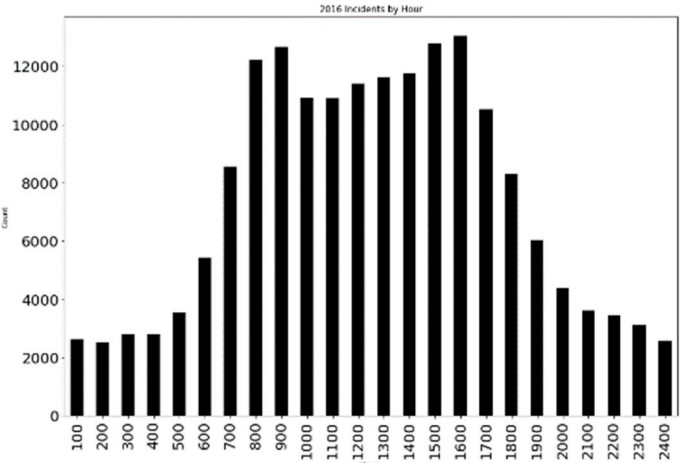

**Figure 5.** 2016 incidents by hour.

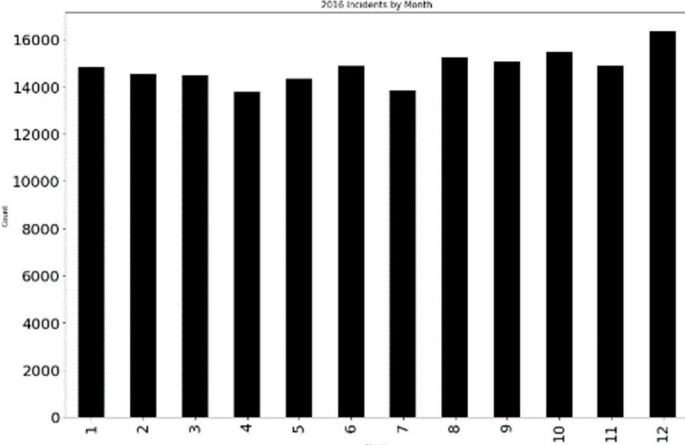

**Figure 6.** 2016 incidents by month.

For this analysis, the trends across the dataset were examined to determine which locations were improving or worsening in incident rates and increasing or decreasing in inspections rates. Figure 8 shows a plot of all counties (using FIPS codes) with the relative slope of the trendline across four years (2016–2019). The same procedure was followed for the inspections for the same four years (2016–2019), which can be seen in Figure 9. In both plots, it is evident that most counties remained steady in both incident and inspection rates, while some experienced extreme improvement/worsening in the case of incidents and increasing/decreasing rates in the case of inspections. The same can be said of the

plot found in Figure 10, which contains a plot of average numbers of incidents across the four years (2016–2019) for all counties (using FIPS codes). Again, most counties remained steady in incident rates, while some increased/decreased dramatically. Lastly, in Figure 11, plots of the slope of the trendline (for years 2016–2019) of the incident-to-inspection ratio, again exhibit the same behavior as the individual plots for incidents and inspections.

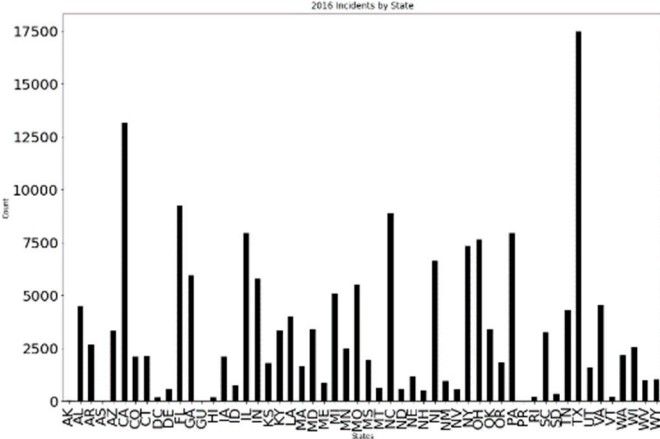

**Figure 7.** 2016 incidents by state.

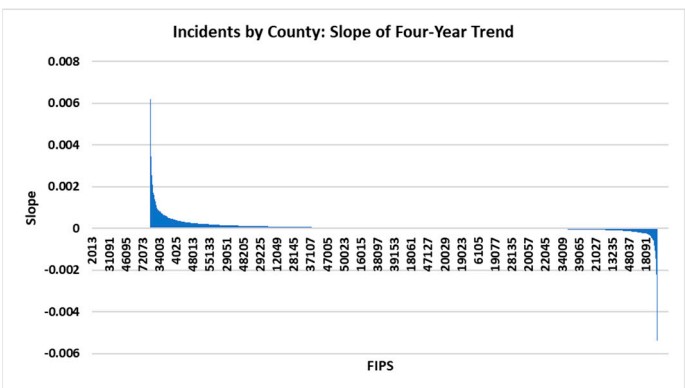

**Figure 8.** Bar plot of four-year trends of incidents across counties (FIPS).

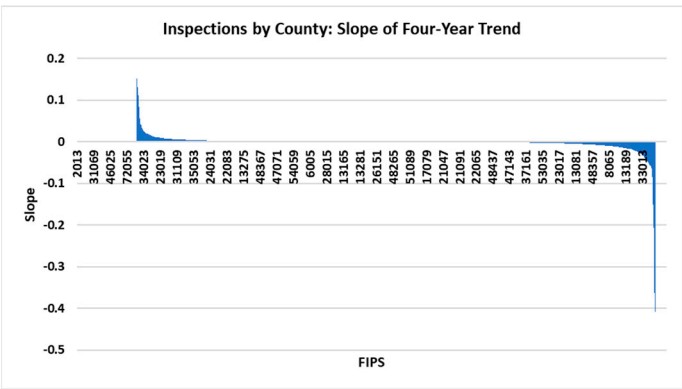

**Figure 9.** Bar plot of four-year trends of inspections across counties (FIPS).

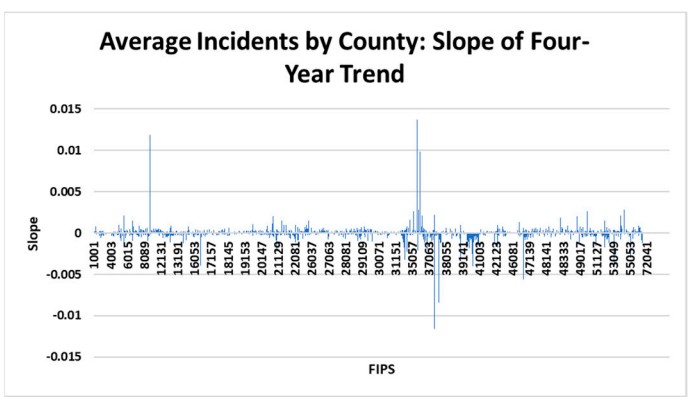

**Figure 10.** Bar plot of four-year trends of average incidents across counties (FIPS).

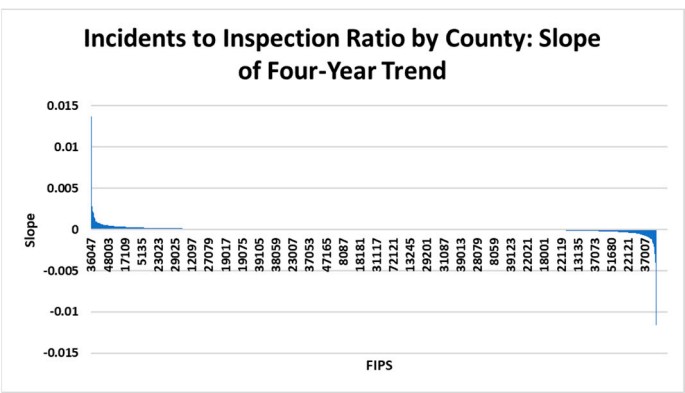

**Figure 11.** Bar plot of four-year trends incidents-to-inspections ratio across counties (FIPS).

Tables 5 and 6 contain the five top and bottom counties from the incident and inspection analyses, which examined the trends across 2016 to 2019. In terms of improvements in incident rates (Table 5), it was surprising that all five were major metropolitan areas, especially Chicago and Phoenix, which have large populations. Total incident rates for the metropolitan areas should be examined to see if a similar trend is found for all vehicular incidents.

**Table 5.** Top Five Counties with Improving and Worsening Incidents Trends (2016–2019).

| | Incidents | | |
|---|---|---|---|
| | **County** | **State** | **County Seat** |
| **Improving** | Oklahoma | Oklahoma | Oklahoma City |
| | Maricopa | Arizona | Phoenix |
| | Tulsa | Oklahoma | Tulsa |
| | Mecklenburg | North Carolina | Charlotte |
| | Cook | Illinois | Chicago |
| **Worsening** | Harris | Texas | Houston |
| | Kings | New York | Brooklyn |
| | Queens | New York | Queens |
| | The Bronx | New York | The Bronx |
| | Bexar | Texas | San Antonio |

**Table 6.** Top Five Counties with Increasing and Decreasing Inspection. Trends (2016–2019).

| | | Inspections | |
|---|---|---|---|
| | **County** | **State** | **County Seat** |
| **Increasing** | El Paso | Texas | El Paso |
| | San Diego | California | San Diego |
| | Webb | Texas | Laredo |
| | Los Angeles | California | Los Angeles |
| | San Bernadino | California | San Bernadino |
| **Decreasing** | Chambers | Texas | Anahuac |
| | Woodruff | Arkansas | Augusta |
| | San Saba | Texas | San Saba |
| | Fayette | Texas | La Grange |
| | Moore | Tennesse | Lynchburg |

In terms of increasing incident rates, it is noteworthy that Houston experienced a significant increase in incidents. This may be due to increased growth in truck traffic in the area, construction, a higher-than-average occurrence of drunk drivers on the road, or other causal variables. An in-depth examination is necessary to pinpoint the cause, and inspection data should be examined to determine if a portion of these incidents could have been prevented. Other than Houston and San Antonio, it is noteworthy that three of the five counties were New York boroughs. Other than congested roadways, parking is an issue in New York, especially for trucks, and further investigation is needed to see if any of the incidents involved trucks circling areas for parking, especially during inclement weather.

In Table 6, for the five counties with increasing inspection rates, it is not surprising to find counties which lie on the border with Mexico (El Paso, San Diego, and Webb). The remaining two counties, Los Angeles and San Bernadino, are relatively close to the border with Mexico and are likely subjected to higher inspection rates. Conversely, in terms of decreasing inspection rates, all five counties listed have extremely low populations (under 3000), except for Lynchburg, Tennessee, which has a population of under 7000. Population rates for these counties should be examined, as well as business and industry growth patterns for these counties and the surrounding areas. It is also noteworthy, that except for Chambers and Fayette counties in Texas, the remaining three counties do not have a major interstate highway (Chambers and Fayette contain segments of I-10). In future analyses, it may be necessary to explore the possibility of closures of truck stops or other locations of DOT truck inspections.

*3.2. Breakdown by County*

A further breakdown of incidents and inspections by county was necessary to examine individual variations throughout each year, and this was performed by normalizing each county by VMT (as seen in Figures 12–15). For each of the five years ranging from 2016 to 2020, the top two counties (or in some cases, the only county) listed in each of the nine boxes can be found in Table 7. Upon examination of the nine boxes, several locations stood out. Obviously, Washington, D.C. stands out, as it appears in Box 9 every year. This is not surprising due to the heavy traffic and maximum security in the area. Strafford County, New Hampshire is also in Box 9 in 2019 and 2020. The Spaulding Turnpike along the Maine border is the primary thoroughfare in that area, thus heavy traffic could be to blame, but further investigation is needed. Ashe County, North Carolina is in Box 8 for three out of the five years. This may be due to the fact that the main highway (US 221) through the county has many turns, which may present an issue to truck drivers. It is not surprising that in 2017 and 2018, Santa Cruz County, Arizona, and in 2019, Otero County, New Mexico are all located in Box 6 (high inspections) as these counties are on

the border with Mexico. Goshen County, Wyoming is in Box 5 in 2016 and 2017. The only notable points of interest in this county are a correctional facility and rodeo fairgrounds, which could both contribute to moderate incidents and inspections. It is noteworthy that for Box 3, in 2016 and 2018, no counties fell within this grouping. Additionally, for 2017, 2019, and 2020, only one county was listed in Box 3. It is noteworthy that the island of Nantucket County, Massachusetts is in Box 7 in 2020. A further investigation is needed to determine the cause of an uptick in incidents; although the lack of inspections on the island is not surprising, especially considering that this was the year of the pandemic. Lastly, it is noteworthy that Texas appeared most frequently throughout the groups. Since all the values were normalized based on truck VMT, this cannot be attributed to simply higher truck volumes in Texas. However, Texas is known to be one of the states with the highest number of drunk-driving-related incidents [16], which could have contributed to higher numbers of truck incidents overall.

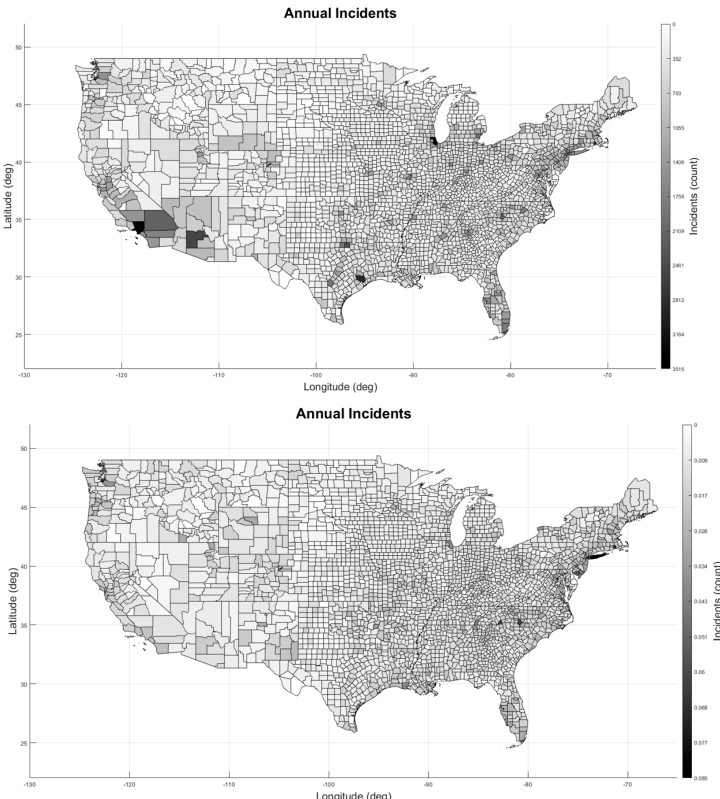

**Figure 12.** Heavy-duty tractor-trailer freight incidents per county in 2016 (absolute value (**top**), normalized by daily VMT (**bottom**)).

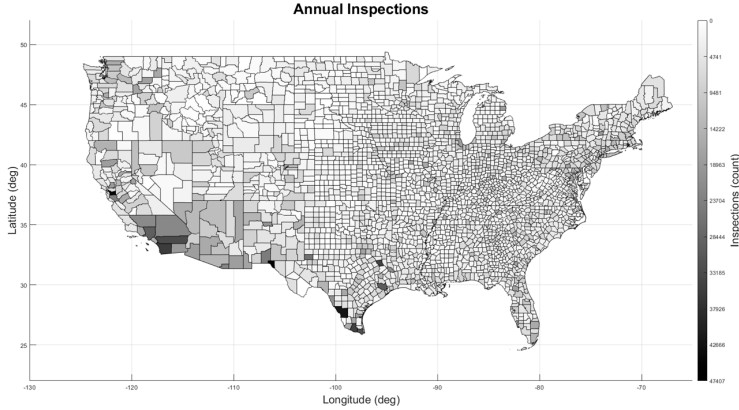

**Figure 13.** *Cont.*

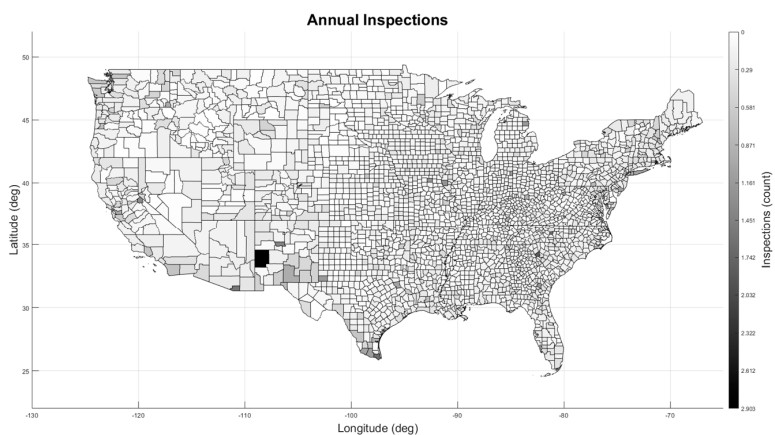

**Figure 13.** Heavy-duty tractor-trailer freight inspections per county in 2016 (absolute value (**top**), normalized by daily VMT (**bottom**)).

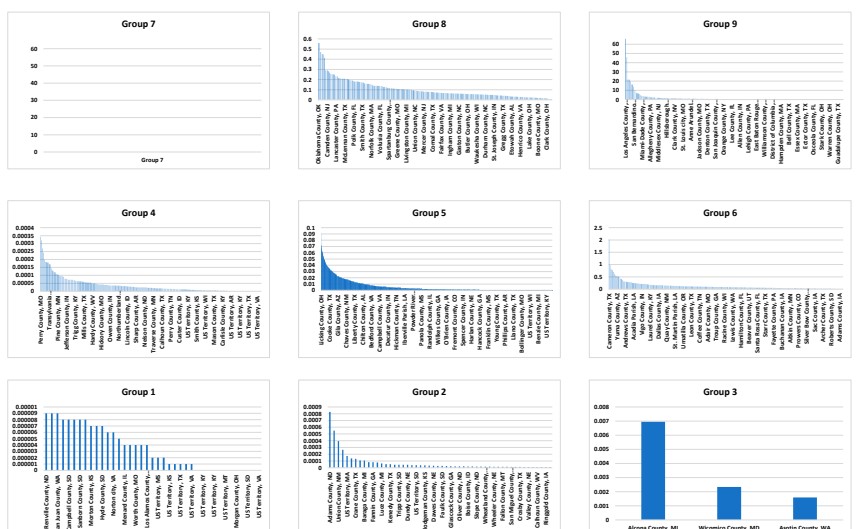

**Figure 14.** Incidents per county in 2016, divided by 1,000,000 and not normalized.

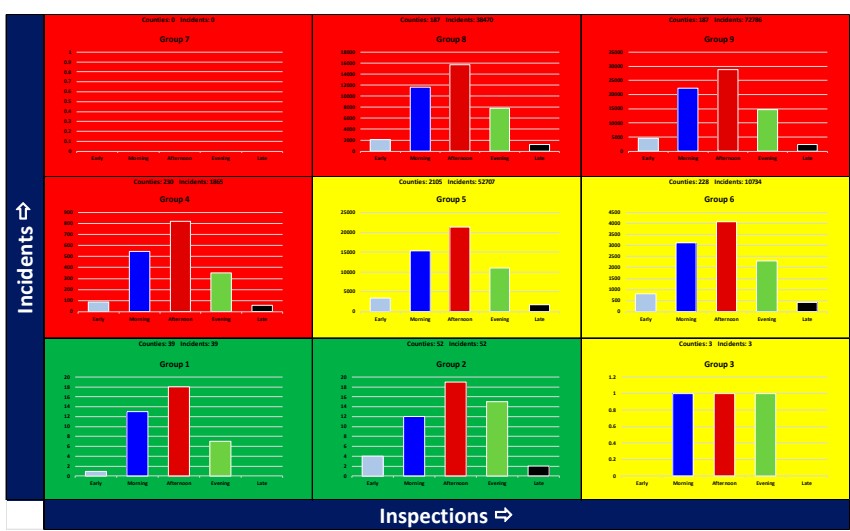

**Figure 15.** Temporal breakdown of absolute number of incidents and inspections in 2016 (aligned with the nine boxes from Figure 4).

**Table 7.** Top two counties in each box.

| Box | Description | Designation | 2016 | 2017 | 2018 | 2019 | 2020 |
|---|---|---|---|---|---|---|---|
| 1 | Low incidents, low inspections | Ideal | Clark (ID), Crosby (TX) | Sedgwick (CO) | None | Jackson (SD), Sedgwick (CO) | Carter (MO), Hall (TX) |
| 2 | Low incidents, moderate inspections | Ideal | Crane (TX), Baraga (MI) | Iron (WI), Pocahontas (IA) | Walworth (SD), Duval (TX) | Choctaw, Pawnee (OK) | Schleicher (TX), Ashland (WI) |
| 3 | Low incidents, high inspections | Less than ideal | None | Republic (KS) | None | Cleveland (OK) | Kinney (TX) |
| 4 | Moderate incidents, low inspections | Not ideal | Casey (KY), Culpepper (VA) | Pendleton (KY), Benewah (ID) | Smith (MS), Lafayette (MO) | Petersburg (VA), Trimble (KY) | Pottawatomie (OK), Perry (OH) |
| 5 | Moderate incidents, moderate inspections | Less than ideal | Baldwin (GA), Goshen (WY) | Goshen (WY), Walthall (MS) | Coffee (AL), Sutter (CA) | Santa Barbara (CA), Rhea (TN) | Floyd (GA), Clay (AL) |
| 6 | Moderate incidents, high inspections | Less than ideal | Abbeville (SC), Adams (IN) | Santa Cruz (AZ), Jackson (MS) | Santa Cruz (AZ), San Diego (CA) | Otero (NM), Stevens (WA) | Clay (NC), Dutchess (NY) |
| 7 | High incidents, low inspections | Not ideal | Alexandria, Petersburg (VA) | Todd (TX), Richmond (VA) | Jackson (MS), Calloway (KY) | Greenup, Hancock (KY) | Calloway (KY), Nantucket (MA) |
| 8 | High incidents, moderate inspections | Not ideal | Mecklenburg (NC), Hudson (NJ) | Ashe (NC), Henry (VA) | Ashe (NC), Tyler (WV) | New York (NY), Cass (MI) | Ashe (NC), Seminole (FL) |
| 9 | High incidents, high inspections | Not ideal | D.C., Alexander (NC) | D.C., Vermillion (LA) | D.C., Cameron (TX) | D.C., Strafford (NH) | D.C., Strafford (NH) |

### 3.3. Temporal Breakdown

When examining the dataset by incident frequency by time of day (Figure 15 and Table 8), it was not surprising that most incidents occurred in the afternoon. This was the assumption, based on previous research which found that most truck incidents occurred between noon and 3:00 p.m. [17]. Out of the five years of data, only four of the boxes (Box 1, two in Box 2, and Box 7) had most incidents taking place in the morning time bin. In all of these instances, inspections were low or moderate. In five of the boxes (Box 1, Box 2, two in Box 3, and Box 7), incidents occurred primarily in the evening. Again, in these cases, inspections were primarily low or moderate. These are not surprising findings, since most inspections take place during typical work hours (9:00 a.m. to 5:00 p.m.).

**Table 8.** Frequency of incidents by time of day.

| Box | Description | Designation | 2016 | 2017 | 2018 | 2019 | 2020 |
|---|---|---|---|---|---|---|---|
| 1 | Low incidents, low inspections | Ideal | Morning | Afternoon | None | Evening | Afternoon |
| 2 | Low incidents, moderate inspections | Ideal | Morning | Afternoon | Evening | Morning | Afternoon |
| 3 | Low incidents, high inspections | Less than ideal | None | Evening | None | Afternoon | Evening |
| 4 | Moderate incidents, low inspections | Not ideal | Afternoon | Afternoon | Afternoon | Afternoon | Afternoon |
| 5 | Moderate incidents, moderate inspections | Less than ideal | Afternoon | Afternoon | Afternoon | Afternoon | Afternoon |
| 6 | Moderate incidents, high inspections | Less than ideal | Afternoon | Afternoon | Afternoon | Afternoon | Afternoon |
| 7 | High incidents, low inspections | Not ideal | Afternoon | Evening | Afternoon | Morning | Afternoon |
| 8 | High incidents, moderate inspections | Not ideal | Afternoon | Afternoon | Afternoon | Afternoon | Afternoon |
| 9 | High incidents, high inspections | Not ideal | Afternoon | Afternoon | Afternoon | Afternoon | Afternoon |

### 3.4. Out-of-Service (OOS) Violations

In Figure 16 and Table 9, it is apparent that the primary OOS violation category was brakes, which was expected. The secondary violation category was tires, which was also

expected. However, this category was found in six cases, and four out of the six occurred in boxes with low inspections and moderate to high levels of incidents. Further investigation is needed, but road conditions in these areas could be considered, especially in construction zones. It is plausible that more incidents caused by tire issues may have been prevented with higher frequencies of inspections in these areas.

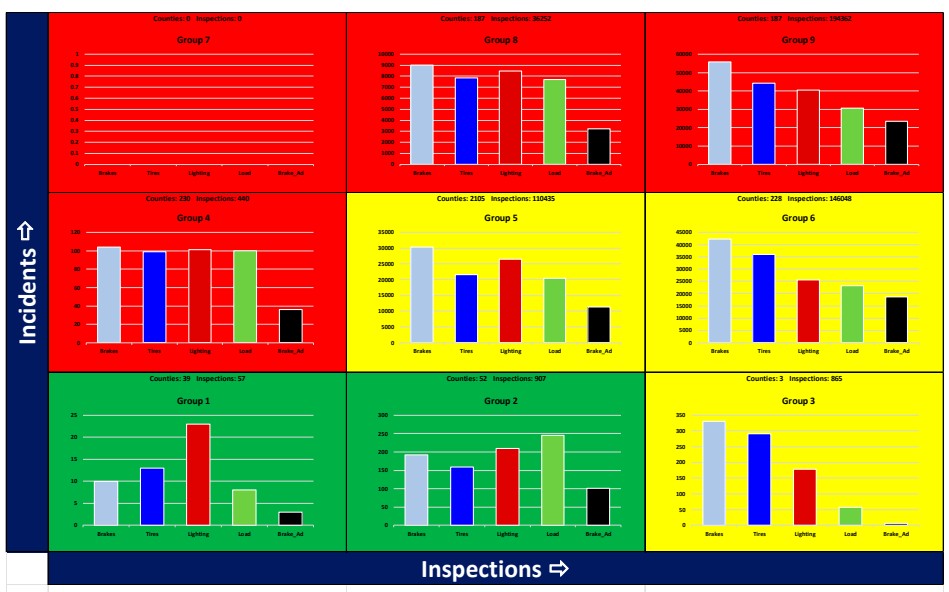

**Figure 16.** Leading OOS inspection recordables associated with incidents and inspections in 2016 (aligned with the nine boxes from Figure 4).

**Table 9.** Incidents by OOS violation category.

| Box | Description | Designation | 2016 | 2017 | 2018 | 2019 | 2020 |
|-----|-------------|-------------|------|------|------|------|------|
| 1 | Low incidents, low inspections | Ideal | Lighting | None | None | Brakes | Lighting |
| 2 | Low incidents, moderate inspections | Ideal | Brakes | Tires | Brakes | Brakes | Brakes |
| 3 | Low incidents, high inspections | Less than ideal | None | Tires | None | Load | Lighting |
| 4 | Moderate incidents, low inspections | Not ideal | Lighting | Brakes | Tires | Brakes | Tires |
| 5 | Moderate incidents, moderate inspections | Less than ideal | Brakes | Brakes | Brakes | Brakes | Brakes |
| 6 | Moderate incidents, high inspections | Less than ideal | Brakes | Brakes | Brakes | Brakes | Brakes |
| 7 | High incidents, low inspections | Not ideal | Lighting | None | Tires | Tires | Lighting |
| 8 | High incidents, moderate inspections | Not ideal | Lighting | Brakes | Brakes | Brakes | Brakes |
| 9 | High incidents, high inspections | Not ideal | Brakes | Brakes | Brakes | Brakes | Brakes |

### 3.5. Non-Out-of-Service (NON-OOS) Violations

In Figure 17 and Table 10, it is apparent that lighting is the primary violation category for the Non-OOS data, which was the assumption based on previous reports. However, vehicle defects, which according to the FMCSA, can include anything from smoke, leaks, or missing/incorrect placards for identifying hazardous materials [17], was the secondary violation category, and is a noticeably more frequent occurrence in 2018, 2019, and 2020. Further investigation is needed, but regulations may have changed during this time, or inspectors may have recategorized an inoperable light as a vehicle defect [18].

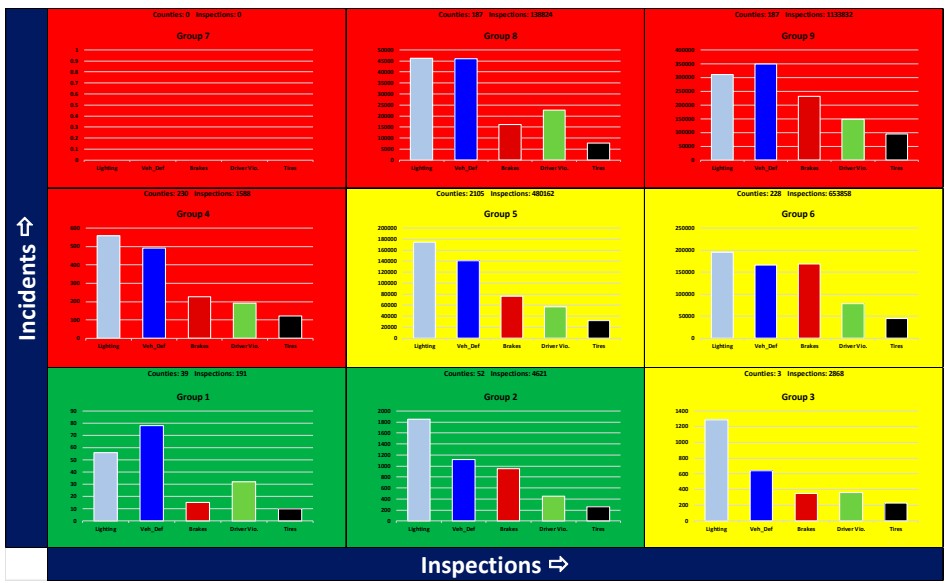

**Figure 17.** Leading Non-OOS inspection recordables associated with incidents and inspections in 2016 (aligned with the nine boxes from Figure 4).

**Table 10.** Incidents by Non-OOS violation category.

| Box | Description | Designation | 2016 | 2017 | 2018 | 2019 | 2020 |
|---|---|---|---|---|---|---|---|
| 1 | Low incidents, low inspections | Ideal | Lighting | Lighting | None | Brakes | Lighting |
| 2 | Low incidents, moderate inspections | Ideal | Lighting | Lighting | Lighting | Vehicle Defects | Lighting |
| 3 | Low incidents, high inspections | Less than ideal | None | Lighting | None | Lighting | Lighting |
| 4 | Moderate incidents, low inspections | Not ideal | Lighting | Lighting | Vehicle Defects | Lighting | Lighting |
| 5 | Moderate incidents, moderate inspections | Less than ideal | Lighting | Lighting | Lighting | Lighting | Lighting |
| 6 | Moderate incidents, high inspections | Less than ideal | Brakes | Lighting | Lighting | Lighting | Lighting |
| 7 | High incidents, low inspections | Not ideal | Lighting | None | Vehicle Defects | Vehicle Defects | Vehicle Defects |
| 8 | High incidents, moderate inspections | Not ideal | Lighting | Lighting | Vehicle Defects | Vehicle Defects | Lighting |
| 9 | High incidents, high inspections | Not ideal | Lighting | Lighting | Vehicle Defects | Vehicle Defects | Vehicle Defects |

## 4. Discussion

### 4.1. Initial Assumptions Regarding Harris, County

As assessed above, Harris County, Texas (Houston) is one of several critical incident centers in the United States. There can be several reasons for a significant number of incidents involving heavy-duty trucks in Houston. Some of the contributing factors may include the following:

1. High traffic volume: Houston is known for its heavy traffic, especially during peak commuting hours. The city's population, economic activity, and extensive transportation infrastructure contribute to the congestion on its roadways. Heavy-duty trucks are often a part of this traffic, and the increased volume can lead to a higher likelihood of incidents.

2. Complex road system: Houston's road system is characterized by numerous highways, interstates, and intricate urban streets. Megaregions with these complex interchange systems and multiple exits, require truck drivers to make frequent lane changes, merge with fast-moving traffic, or navigate unfamiliar routes. Maneuvering large

trucks through these intricate interchanges can be challenging, leading to errors or misjudgments that contribute to incidents. As such, the complexity of the road system increases the risk of incidents involving heavy-duty trucks.

3. Interconnected transportation systems: Houston is part of the Texas Triangle Megaregion. This megaregion encompasses five of the largest 20 U.S. cities and is home to more than 70% of Texans—a population of nearly 21 million people. This region is formed by the state's four main urban centers, Austin, Dallas–Fort Worth, Houston, and San Antonio, connected by Interstate 45, Interstate 10, and Interstate 35. The Texas Triangle is one of the country's eleven megaregions, which are clusters of urban areas that share economic and cultural ties. This region experiences 306 MT of daily truck freight movement, or 5.3% of the total U.S. truck freight movement, through an average of ~35.7 k miles of daily commercial VMT (see Figure 18). The interconnectivity of these transportation networks means that trucks are commonly involved in long-haul trips, intercity deliveries, or distribution activities. The extensive truck movement across different regions and routes can increase the exposure to potential incident risks.

4. Interaction with vulnerable road users: Megaregions typically have a higher concentration of pedestrians, cyclists, and other vulnerable road users. The increased interaction between trucks and these users can raise the risk of incidents, especially at intersections, crosswalks, or areas with heavy pedestrian activity.

5. Driver fatigue: Truck drivers often work long hours and face demanding schedules. The pressure to meet delivery deadlines can lead to fatigue and drowsiness. Fatigue impairs a driver's cognitive abilities and reaction times, making it more difficult to maintain focus and respond effectively to changing road conditions. Fatigued truck drivers are more prone to incidents.

6. Weather conditions: Houston experiences a range of weather conditions, including heavy rainfall, fog, and occasional severe storms. These weather events can reduce visibility, create slippery road surfaces, and cause hydroplaning. Heavy-duty trucks, due to their size and weight, require additional stopping distance and maneuvering capabilities, making them more susceptible to incidents during adverse weather conditions.

7. Inadequate training: Safe operation of heavy-duty trucks requires specialized skills and knowledge. If truck drivers are not adequately trained in handling these large vehicles, understanding safety protocols, or responding to various scenarios, it can increase the risk of incidents. Insufficient training may result in errors in judgment, improper vehicle handling, or a lack of awareness of blind spots, contributing to incidents.

8. Maintenance and mechanical issues: Mechanical failures in heavy-duty trucks can occur due to poor maintenance practices or faulty equipment. Brake malfunctions, tire blowouts, steering problems, or engine issues can significantly impact a truck driver's ability to control the vehicle safely. Failure to address or detect these mechanical issues in a timely manner can lead to incidents.

9. Unsafe driving practices: Some incidents involving heavy-duty trucks in Houston can be attributed to unsafe driving practices. Speeding, tailgating, improper lane changes, distracted driving (such as using mobile devices), or driving under the influence of alcohol or drugs are examples of behaviors that increase the risk of incidents. These unsafe practices can endanger not only the truck driver but also other road users.

Addressing these factors, especially in megaregions such as Houston, requires a comprehensive approach involving infrastructure improvements, driver education and training, improved traffic management, enhanced maintenance programs, stricter enforcement of safety regulations, and public awareness programs. By promoting safety awareness and implementing measures to mitigate these risks, it is possible to reduce the number of incidents involving heavy-duty trucks in regions such as Houston. It is important to note that the specific causes and factors contributing to incidents involving heavy-duty trucks

in Houston can vary on a case-by-case basis. Detailed incident investigations, conducted by law enforcement authorities and transportation agencies, can provide more specific insights into the causes of individual incidents.

| Year | Type | Truck Kton-miles (Annual) | | | Truck VMT (Daily) | | |
|---|---|---|---|---|---|---|---|
| | | Region | US | % Share | Region | US | Regional share |
| 2017 | Single | 28,533,150 | 560,873,845 | 5.09% | 4,175,820 | 88,740,317 | 4.71% |
| 2017 | Combination | 83,122,180 | 1,538,794,017 | 5.40% | 8,853,270 | 172,422,852 | 5.13% |
| **2017** | **Total** | **111,655,327** | **2,099,667,813** | **5.32%** | **13,029,089** | **261,163,164** | **4.99%** |

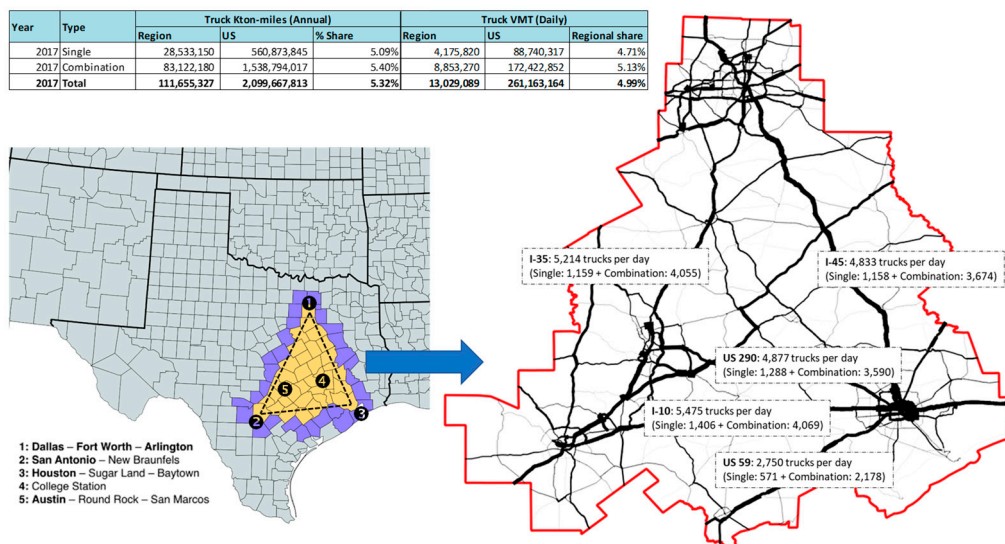

**Figure 18.** Houston, Texas and the Texas Triangle Megaregion.

*4.2. Initial Assumptions Regarding New York Boroughs*

It is noteworthy that out of the top five U.S. counties with increasing truck accident rates during the 2016–2019 study period, after Harris County, Texas (Houston), the following top three counties are all New York boroughs (Kings, Queens, and the Bronx). As New York City is the most heavily populated city in the U.S. and is the most densely populated, it is not completely surprising that incidents would occur. However, it is significant that three of the five New York boroughs had upward trends in incident rates during the study period. Initially, this could be attributed to several factors, which may include the following:

1. Interaction with vulnerable road users: New York is the most densely populated city in the U.S. It also has one of the most expansive public transit networks in the world. Due to its sprawling network of heavy/light rail, commuter rail, and buses, many residents of New York are less car-dependent than in other U.S. city. Although this is great for many reasons, it also unfortunately provides an opportunity for more pedestrian vehicle incidents, and incidents caused by pedestrians and cyclists, especially for medium- and heavy-duty trucks.

2. Unsafe driving practices: New York is known for its ubiquitous taxi fleets. With the introduction of ride-hailing services such as Uber and Lyft, this introduces even more vehicles on the streets, diving in and out of parking lanes and shoulders. The presence of food delivery services (Uber Eats, GrubHub, and the like) also adds to the often-times chaotic traffic scenes, typical of New York streets.

3. Complex road system: A popular focus of the freight and transportation research being conducted at universities in New York (including, but not limited to Rensselaer Polytechnic Institute and the City College of New York) include the issue of a lack of on-street parking for delivery vehicles, and the occurrence of parking violations, off-hour delivery restrictions, and other means of managing the lack of curb space within the boroughs. This presents unique challenges for medium- and heavy-duty trucks.

4. Interconnected transportation systems: The boroughs of New York are known for their massive, multilane roadways [19–21] (Figure 19) and some of the country's first parkways (Eastern Parkway, Bronx River Parkway, etc.). The notorious Cross Bronx Expressway (part of I-95) likely comes to mind, with its typical traffic jams and frequent incidents. These complicated stretches of highway are part of an interconnected network within the boroughs, and are often at a standstill during peak

hours, but are also the location of many incidents due to lane changes and frequency of ingress/egress points.

5. Complexities due to ongoing construction: As the major thoroughfares in New York are subjected to vast amounts of vehicle traffic daily, as well as frequent and severe weather events, many of the roadways are constantly under construction. As of 2023, the Bruckner Expressway, a major thoroughfare in the Bronx, is scheduled to begin a complete revitalization, and like many of the aging roadways in the state, it is crumbling due to extreme wear and tear. This will ultimately lead to congestion and rerouting of vehicles to nearby roadways, which may result in frequent incidents, and further degradation of New York's aging roadway network.

6. Weather conditions: The state of New York has experienced many severe weather events in the last few years, ranging from blizzards, Nor'Easters, hurricanes, and flash floods, all leading to traffic events, and even unforeseen damage to bridges and roadways. Unaddressed damage caused by severe weather, even something as simple as repairing potholes, can lead to significant traffic events and incidents, especially for heavy-duty trucks.

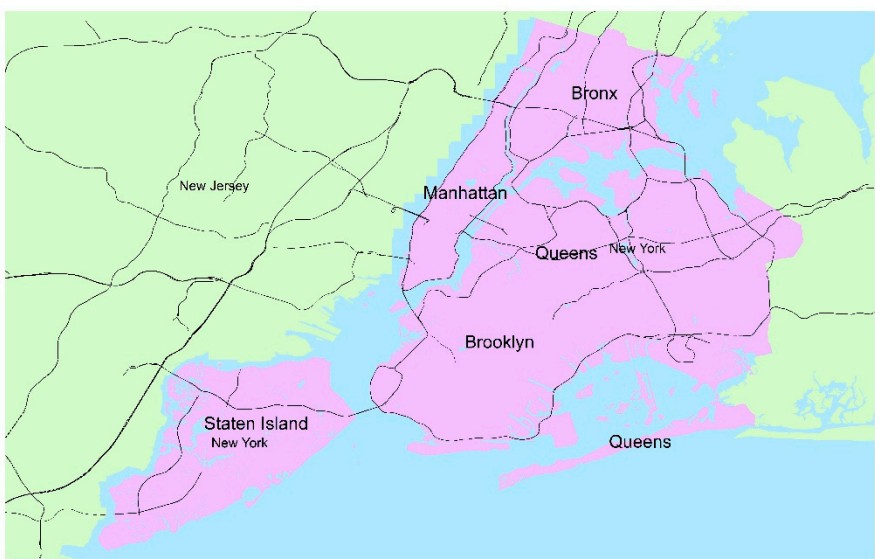

**Figure 19.** New York boroughs with primary roads.

Again, these factors, as well as many others, must be addressed to uncover the underlying causes of an increase in incidents in the New York boroughs. A comprehensive approach is certainly needed to address and anticipate future infrastructure improvements and maintenance, improve and increase driver training, improve traffic management strategies, improve enforcement of safety regulations, and public education programs. As with Houston, efforts to increase safety awareness and implementing measures to mitigate these risks will make it possible to reduce the number of incidents involving heavy-duty trucks throughout New York and the New York boroughs. Additionally, as with Houston, specific causes and factors contributing to incidents involving heavy-duty trucks in the New York boroughs certainly vary on a case-by-case basis, and detailed investigations will further provide more specific insights into the causes of individual incidents.

*4.3. Recommendations on Enforcement of Safety*

A major goal for every fleet is to reduce the number and severity of incidents. Professional safety training is key in achieving this goal. Although safety is typically a stated goal of any fleet, demonstrating this through consistent safety training is crucial. Management of driver behavior is also important [22], and this can be performed through the use of on-board management systems (OBMS), which monitor the driver. Understandably, there is opposition to the use of these systems, as drivers typically feel a level of discomfort in

being monitored throughout their duration on the road. However, these systems have proven to be excellent in not only providing justification in legal situations, where the truck driver was falsely accused of a traffic violation, but they have also proven to be appropriate devices for teaching safe driving skills. Just as importantly, these systems provide real-time telematics data for drivers and fleet managers, which provides a means of being proactive in avoiding incidents due to imminent vehicle failures, thus increasing operator awareness [23]. Maintaining consistency across the board, not only with training and vehicle maintenance, has been shown to reduce the number and severity of incidents. Fleets with consistent routes, scheduling, and fleet personnel, with low turnover rates, have all shown reduced numbers of incidents [24]. Additional studies may benefit from further examination of proactive measures taken by fleet managers, which correlate with decreasing incidents.

While the aforementioned practices are strategies that fleet managers can implement, there are certainly opportunities from a regulatory perspective that can be implemented to ensure greater safety on the road. According to 49 CFR 396: Minimum Periodic Inspection Standards [24], vehicles are required to undergo inspection at least once within a 12-month period. However, inspection requirements vary state-to-state, and obviously inspections are costly (in terms of labor and resources, as well as time lost for the carrier). Future research is needed to determine if inspections may be happening too infrequently, and whether the timeline may need to be reconsidered. Currently, a vehicle that has undergone inspection, resulting in an out-of-service (OOS) violation(s), has 15 days to complete repairs and return documentation stating that the cause(s) of the vehicle to fail inspection were addressed [24]. Approximately two weeks may seem like an eternity for someone to be unable to work, which may necessitate the need to obtain parts and labor to address the issues outlined in the OOS report as soon as possible, but it may not be adequate time to appropriately fix the vehicle. This timeline for repairs may need to be reconsidered based on labor and part availability as well as criticality regarding further safe operation of the vehicle. According to 49 CFR 396.11, drivers must also complete and sign a post-trip form, confirming that the vehicle is in adequate condition concerning the following parts [24]:

- Service brakes, including trailer brake connections;
- Parking (hand) brake;
- Steering mechanism;
- Lighting devices and reflectors;
- Tires;
- Horn;
- Windshield wipers;
- Rear-vision mirrors;
- Coupling devices;
- Wheels and rims;
- Emergency equipment.

Appropriate and consistent training is required to ensure that this responsibility of the driver is not taken lightly. The driver (and the carrier) is also responsible for reporting if a repair required following an inspection has been addressed, and it is at their discretion to state whether they believe the repair to be unnecessary. This may allow the opportunity to forego even minor repairs that may result in major issues, perhaps even resulting in an incident later on. All of these aspects regarding the procedure(s) related to regular inspection of the vehicle and the responsibilities of the driver should be re-evaluated annually. Detailed data collection, including further driver surveys, will likely prove beneficial in potentially refining some of these safety regulations and should be considered in future studies.

## 5. Conclusions and Next Steps

The purpose of this work was to take an in-depth look into the FMCSA's MCMIS dataset and attempt to uncover hints of causal variables resulting in truck incidents through-

out the U.S. to increase safety and improve efficiency, both in terms of day-to-day operations and resource management, and in terms of energy usage. This data exploration is novel and has not been performed previously at this level of granularity. Previously, studies have only focused on one state or county, whereas this study provided a national-scale evaluation of the MCMIS dataset. Examining the data at the county level served to pinpoint locations of interest for further inquiry. Performing these analyses at this level of granularity allowed for a more in-depth look at specific locations, which differed from all previous studies using the MCMIS dataset. Temporal categorization of the data provided support of previous research and may also lead to support for expansion of operating hours for DOT inspections. The breakdown by OOS and Non-OOS categories provided an opportunity to examine spatio-temporal patterns within the dataset. Examination of the incident data relative to inspections at each county, and evaluation of the data using an incident-to-inspection ratio allowed for an additional level of analysis to explore relative heterogeneity within the dataset. The findings from the analyses support future expansions of this work, with the inclusion of other potential variables including, but not limited to, effects of sun angle on drivers, proximity to inspection facilities, driver characteristics, and the inclusion of improvements in technologies, namely the installation of camera-based mirror systems (CBMS) and varying levels of automation, as well as further ways to reduce fleet-level energy usage and improve energy efficiency from a systems-level perspective.

**Author Contributions:** Conceptualization, V.S., A.M., A.S., H.L., S.O. and S.T.; data curation, V.S., A.S., A.M. and H.L.; formal analysis, V.S. and A.M.; funding acquisition, A.S.; investigation, A.M., V.S. and A.S.; methodology, V.S., A.M., A.S. and H.L.; project administration, V.S.; resources, V.S., A.M., A.S. and H.L.; software, V.S., A.M., A.S. and H.L.; supervision, V.S.; validation, V.S., A.M., A.S. and H.L.; visualization, A.M. and V.S.; writing—original draft, A.M. and V.S.; writing—review and editing, V.S. and A.M. All authors have read and agreed to the published version of the manuscript.

**Funding:** This manuscript has been authored by UT-Battelle, LLC, under contract DE-AC05-00OR22725 with the US Department of Energy (DOE). The US government retains and the publisher, by accepting the article for publication, acknowledges that the US government retains a nonexclusive, paid-up, irrevocable, worldwide license to publish or reproduce the published form of this manuscript, or allow others to do so, for US government purposes. DOE will provide public access to these results of federally sponsored research in accordance with the DOE Public Access Plan (https://www.energy.gov/doe-public-access-plan access on 15 July 2023).

**Institutional Review Board Statement:** This study was conducted in accordance with the Declaration of Helsinki and was approved by the Institutional Review Board (or Ethics Committee) of the Oak Ridge National Laboratory (45CFR46/10CFR745, 10 August 2023).

**Informed Consent Statement:** Not applicable.

**Data Availability Statement:** The data presented in this study are available on request from the corresponding author. All data generated and used in this study are summarized and referenced in the text of the document.

**Conflicts of Interest:** The authors declare no conflict of interest.

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
