# Peer review of "Spatio-Temporal Assessment of Heavy-Duty Truck Incident and Inspection Data"

_safety, 2023_

Round 1

Reviewer 1 Report

The topic and analysis is quite interesting, data are presented clearly, method of calculation is mostly described in the proper way.   Remarks:. In the introduction there is no review of other papers about this topic (state of art). Doesn't exists? It should be provided.   Fig 13 too small description, not readable   Fig. 4 right - how exactly it was normalized (calculated)? In text it was stated that " Additional scatter plots also showed the incident to inspection ratio, relative to inspections for each county"   All data with incidents and inspections numbers should be normalized by VMT, otherwise it is obvious (more/ trucks, higher traffic, more accidents)   Data with incident/inspection ratio are already normalised (more or less) by inspection number (more trucks)   Line 492. "This time line for repairs may need to be reconsidered based on labor and part availability" - conclusion maybe correct, but not supported by any data in the paper not supported by data. Conclusion in scientific papers should be based on presented data   Line 464:  "on-board management systems (OBMS) correlation with accidents? Not supported by data" - the same, not supported by any data in the paper. It can be valuable is the number of used OBMS will be compared with number of accidents   Lookin at the content of the paper, the discussion/conclusions should be maybe rather about recommendation of adjustment of the inspection rate by the usage of presented data.    

Author Response

Dear Reviewer,

Please find below your comments and suggestions for our paper along with our responses. 

Thank you so much.

Remarks:

  • In the introduction there is no review of other papers about this topic (state of art). Doesn't exists? It should be provided.  

Response: Although the actual methods used in this research are not completely novel, the analysis of the MCMIS dataset at the national level hadn’t been done previously. This detail was added in the introduction. County and state-level studies using the MCMIS dataset have been added to the introduction and references. International sources have also been added to justify the need to examine truck maintenance relative to truck incidents.

  • Fig 13 too small description, not readable

Response: The figures have been enlarged.

  • 4 right - how exactly it was normalized (calculated)? In text it was stated that " Additional scatter plots also showed the incident to inspection ratio, relative to inspections for each county".  All data with incidents and inspections numbers should be normalized by VMT, otherwise it is obvious (more/ trucks, higher traffic, more accidents). Data with incident/inspection ratio are already normalized (more or less) by inspection number (more trucks)

Response: All of the analyses used normalized values for incidents and inspections, which were obtained by dividing totals by the VMT. The plot on the left (in Figure 4) was provided to highlight the importance of normalizing based on VMT to prevent locations with higher truck volumes from being overrepresented.

  • Line 492. "This timeline for repairs may need to be reconsidered based on labor and part availability" - conclusion maybe correct, but not supported by any data in the paper not supported by data. Conclusion in scientific papers should be based on presented data.  

Response: This statement was reworded to reflect the need for future research.

  • Line 464:  "on-board management systems (OBMS) correlation with accidents? Not supported by data" - the same, not supported by any data in the paper. It can be valuable is the number of used OBMS will be compared with number of accidents.

Response: Unfortunately, in the MCMIS dataset, the use of OBMS was not provided. The statement regarding OBMS and improved safety is relative to driver behavior management. A study by UC Berkely was referenced.

  • Looking at the content of the paper, the discussion/conclusions should be maybe rather about recommendation of adjustment of the inspection rate by the usage of presented data.  

Response: Thank you for the suggestion. 

Reviewer 2 Report

Thanks for submitting this paper for being considered in Safety. The manuscript (safety-2583152) is empirical study investigating spatio-temporal data on road accidents including heavy goods vehicles. The topic addressed is worth of investigation, and the theoretical/empirical approaches considered result interesting and may contribute in a substantial manner to prioritize road safety measures, particularly in terms of fleet management. The manuscript is easy to follow, the statistically analysis is reasonable, the conclusion is accurate.

 My comments are listed below:

 1.     Although the authors conducted exhaustive literature review, it is not entirely clear to this reviewer how the current study makes a significant contribution above other papers. The authors should be more clear about the relevance of their research for the problem of identifying the spatial and temporal characteristics of road traffic accidents. In my opinion, the introduction may be further clarified and strengthened if the authors describe in a more structured way in what respects their research replicates earlier research. This is one issue that the authors should describe more clearly in the introduction (and in the discussion). I believe that the introduction and discussion should be written more to the point.

Author Response

Dear Reviewer,

Please find below your comments/suggestions for our paper along with our responses.

Thank you so much.

Remarks:

  • Although the authors conducted exhaustive literature review, it is not entirely clear to this reviewer how the current study makes a significant contribution above other papers. The authors should be more clear about the relevance of their research for the problem of identifying the spatial and temporal characteristics of road traffic accidents. In my opinion, the introduction may be further clarified and strengthened if the authors describe in a more structured way in what respects their research replicates earlier research. This is one issue that the authors should describe more clearly in the introduction (and in the discussion). I believe that the introduction and discussion should be written more to the point.

Response: Although the actual methods used in this research are not completely novel, the analysis of the MCMIS dataset at the national level hadn’t been done previously. This detail was added in the introduction. County and state-level studies using the MCMIS dataset have been added to the introduction and references. International sources have also been added to justify the need to examine truck maintenance relative to truck incidents.

Reviewer 3 Report

Dear Authors,

Below, the Authors will find several observations of the manuscript. This reviewer considers that these observations will help to generate a complete document. Moreover, please bear in mind that one of the purposes of a scientific paper is that the research carried out is replicable by other researchers. Please send back a new version of the paper and a response letter indicating how the Authors attended each observation:

1. The flow of the abstract should be the background, the objective of the research, the methodology, the main results, and the main conclusion. The abstract should be rewritten following the requested flow. In the current version of the document, the abstract needs to include the principal results and conclusion. The following sentence is not relevant in the abstract: “This preliminary work is part of an on-going effort funded by FMCSA and serves as the basis for future research to improve vehicle operational safety and increase energy efficiency for all vehicles on the road”; instead of this sentence, please focus on the main results and the main conclusion.

2. Please change the keywords already included in the title to increase the possibility of searching and finding the manuscript.

3. This reviewer thinks the introduction's final part takes the manuscript's results for granted rather than stating the research gap, research questions, and hypotheses. Please consider rewriting the last paragraph of the introduction according to the expected structure of a scientific article.

4. Please provide detailed information on how the Authors established the thresholds for each of the nine boxes in Figure 4. Furthermore, what are the criteria for considering that the incident/inspection ratio is low, moderate, or high in terms of the possible quantitative values of this ratio? This reviewer understands that the information presented in Lines 158 to 167 corresponds to the assumptions to generate Table 4; however, the question is about the quantitative thresholds in Figure 4.

5. This reviewer has several concerns about this manuscript. Although the work carried out has its merits, the manuscript does not meet the characteristics of a scientific article. Please remember that the Safety Journal is an international journal; therefore, the focus of published articles must be international. The case study in the USA is valid; however, the literature review must be international. Following the introduction, please provide a new "literature review" section presenting all relevant international research on assessing heavy-duty truck incidents and inspection data for at least the last five years. This literature review will allow the authors to discuss their results in front of international references and position their findings within the state of knowledge.

6. Consistent with comment 4, the Authors need to make their methodological contribution to the state of knowledge evident. By relegating the method to subsection 2.1, the Authors need to give more recognition to their contributions. This reviewer suggests that a new section titled "Methods" be created so that Authors can explain in detail their method and how it goes beyond what other authors do internationally.

7. The manuscript presents a satisfactory results analysis. However, the discussion section does not correspond to the discussion of the results of a scientific article. According to comment 5, once the manuscript has an updated literature review, it will be possible for the Authors to discuss (compare) their findings against the results of international studies. For example, part of that discussion could focus on whether the 15 factors (nine from Houston and six from New York Boroughs) match or differ from the factors that are determinants in international studies and why. Please note that the different methods used in the studies enrich the discussion; hence, the requested literature review is necessary.

8. The conclusions focus on supporting the need for this preliminary study, leaving aside the relevant conclusions that correspond to the study findings and how these findings can help internationally, improve inspection practices, and reduce the number of incidents. The conclusions of a scientific article must return to the research questions and hypotheses (established in the introduction) and answer these questions according to the results, as well as accept or not the hypotheses arguing why. This reviewer invites the authors to restate their conclusions, giving relevance to their findings, which are their product to show.

kind regards,

The reviewer

Author Response

Dear Reviewer,

Please find below your comments/suggestions for our paper along with our responses. 

Thank you so much.

  1. The flow of the abstract should be the background, the objective of the research, the methodology, the main results, and the main conclusion. The abstract should be rewritten following the requested flow. In the current version of the document, the abstract needs to include the principal results and conclusion. The following sentence is not relevant in the abstract: “This preliminary work is part of an on-going effort funded by FMCSA and serves as the basis for future research to improve vehicle operational safety and increase energy efficiency for all vehicles on the road”; instead of this sentence, please focus on the main results and the main conclusion.

Response: The statement regarding the funding source from FMCSA has been removed and the abstract has been heavily edited to reflect your suggestions.

  1. Please change the keywords already included in the title to increase the possibility of searching and finding the manuscript.

Response: The keywords have been altered.

  1. This reviewer thinks the introduction's final part takes the manuscript's results for granted rather than stating the research gap, research questions, and hypotheses. Please consider rewriting the last paragraph of the introduction according to the expected structure of a scientific article.

Response: This paragraph has been edited to address these concerns.

  1. Please provide detailed information on how the Authors established the thresholds for each of the nine boxes in Figure 4. Furthermore, what are the criteria for considering that the incident/inspection ratio is low, moderate, or high in terms of the possible quantitative values of this ratio? This reviewer understands that the information presented in Lines 158 to 167 corresponds to the assumptions to generate Table 4; however, the question is about the quantitative thresholds in Figure 4.

Response: The nine-box designation was arbitrary. The low, moderate, and high designations were simply based on the box arrangement. The assumptions were arbitrary and not based on prior research. The authors were attempting to create a visual classification system for the plotted incidents and inspections.

  1. This reviewer has several concerns about this manuscript. Although the work carried out has its merits, the manuscript does not meet the characteristics of a scientific article. Please remember that the Safety Journal is an international journal; therefore, the focus of published articles must be international. The case study in the USA is valid; however, the literature review must be international. Following the introduction, please provide a new "literature review" section presenting all relevant international research on assessing heavy-duty truck incidents and inspection data for at least the last five years. This literature review will allow the authors to discuss their results in front of international references and position their findings within the state of knowledge.

Response: Although an additional literature review section hasn’t been added, several international studies have been added to the introduction and references. These studies support the statement that vehicle maintenance affects incident frequency and that vehicular incidents, especially those involving heavy-duty trucks, are on the rise worldwide. Thus, requiring a focus on prevention strategies, namely in terms of improving vehicle inspection and vehicle maintenance.  

  1. Consistent with comment 4, the Authors need to make their methodological contribution to the state of knowledge evident. By relegating the method to subsection 2.1, the Authors need to give more recognition to their contributions. This reviewer suggests that a new section titled "Methods" be created so that Authors can explain in detail their method and how it goes beyond what other authors do internationally.

Response: The introduction and conclusion have been edited to recognize the contribution of this work and highlight how these analyses differ from previous studies using the same, or similar datasets.

  1. The manuscript presents a satisfactory results analysis. However, the discussion section does not correspond to the discussion of the results of a scientific article. According to comment 5, once the manuscript has an updated literature review, it will be possible for the Authors to discuss (compare) their findings against the results of international studies. For example, part of that discussion could focus on whether the 15 factors (nine from Houston and six from New York Boroughs) match or differ from the factors that are determinants in international studies and why. Please note that the different methods used in the studies enrich the discussion; hence, the requested literature review is necessary.

Response: The focus was on the use of a US dataset (MCMIS), which was why international studies were originally excluded from comparison. Additional sources citing use of the MCMIS dataset with less granularity have been included. International sources have also been included and referenced in the introduction.

  1. The conclusions focus on supporting the need for this preliminary study, leaving aside the relevant conclusions that correspond to the study findings and how these findings can help internationally, improve inspection practices, and reduce the number of incidents. The conclusions of a scientific article must return to the research questions and hypotheses (established in the introduction) and answer these questions according to the results, as well as accept or not the hypotheses arguing why. This reviewer invites the authors to restate their conclusions, giving relevance to their findings, which are their product to show.

Response: The conclusion has been edited to reflect the relevance of this study and the significance of the findings.

Round 2

Reviewer 1 Report

The paper can be probably accepted, but in pdf opened by Adobe Reader many references has:

"Error! Reference source not found"

Reviewer 2 Report

I have no additional comments.

Reviewer 3 Report

The authors have satisfactorily responded to the requests, achieving a better presentation of their research.